# When has service provision for transient ischaemic attack improved enough? A discrete event simulation economic modelling study

Pelham Barton,[1] James P Sheppard,[2] Cristina M Penaloza-Ramos,[1] Sue Jowett,[1] Gary A Ford,[3] Daniel Lasserson,[4] Jonathan Mant,[5] Ruth M Mellor,[6] Tom Quinn,[7] Peter M Rothwell,[8] David Sandler,[9] Don Sims,[10] Richard J McManus,[2] on behalf of the BBC CLAHRC investigators

For numbered affiliations see end of article.

**Correspondence to**
Dr James P Sheppard;
james.sheppard@phc.ox.ac.uk

## ABSTRACT

**Objectives** The aim of this study was to examine the impact of transient ischaemic attack (TIA) service modification in two hospitals on costs and clinical outcomes.

**Design** Discrete event simulation model using data from routine electronic health records from 2011.

**Participants** Patients with suspected TIA were followed from symptom onset to presentation, referral to specialist clinics, treatment and subsequent stroke.

**Interventions** Included existing versus previous (less same day clinics) and hypothetical service reconfiguration (7-day service with less availability of clinics per day).

**Outcome measures** The primary outcome of the model was the prevalence of major stroke after TIA. Secondary outcomes included service costs (including those of treating subsequent stroke) and time to treatment and attainment of national targets for service provision (proportion of high-risk patients (according to ABCD$^2$ score) seen within 24 hours).

**Results** The estimated costs of previous service provision for 490 patients (aged 74±12 years, 48.9% female and 23.6% high risk) per year at each site were £340 000 and £368 000, respectively. This resulted in 31% of high-risk patients seen within 24 hours of referral (47/150) with a median time from referral to clinic attendance/treatment of 1.15 days (IQR 0.93–2.88). The costs associated with the existing and hypothetical services decreased by £5000 at one site and increased £21 000 at the other site. Target attainment was improved to 79% (118/150). However, the median time to clinic attendance was only reduced to 0.85 days (IQR 0.17–0.99) and thus no appreciable impact on the modelled incidence of major stroke was observed (10.7 per year, 99% CI 10.5 to 10.9 (previous service) vs 10.6 per year, 99% CI 10.4 to 10.8 (existing service)).

**Conclusions** Reconfiguration of services for TIA is effective at increasing target attainment, but in services which are already working efficiently (treating patients within 1–2 days), it has little estimated impact on clinical outcomes and increased investment may not be worthwhile.

## Strengths and limitations of this study

► This study focused on two hospitals within the West Midlands, UK, so the results are likely to be representative of these hospitals, but not necessarily other centres in the UK and across the world.
► Modelling allows estimation of service 'unknowns' which requires preprogramming and therefore simplification of certain service intricacies. Thus, the services modelled represent stylised versions of the actual services offered by participating hospitals.
► Costs relate to standard weekday services and it is not clear whether those actually incurred would be increased by provision of services at the weekend.
► The potential costs of implementation of new services, for example those for weekend services, were unavailable and so the costs presented here may have been underestimated.

## BACKGROUND

Transient ischaemic attack (TIA) is common, with incidences of 15–83 patients per 100 000 population recorded across the world[1–5]; in the UK, it affects approximately 50 patients per 100 000 population.[6] TIA is important because it represents a significant risk factor for future stroke with around 8% suffering an event within 7 days, and 12% within a month without preventative therapy.[7 8]

Rapid recognition and treatment of TIA patients at high risk of having a subsequent stroke is important because simple interventions, such as early prescription of preventative medications, can substantially reduce the risk of stroke following TIA, and existing evidence suggests that earlier intervention may be better.[9 10] In the UK, guidelines[11 12] recommend that patients at high risk of recurrent stroke (defined using the ABCD$^2$ score of >4)[10] are seen within 24 hours of symptom onset and all other patients are seen within 7 days. A recent UK audit suggests that there has been significant improvement

in attainment of these targets: 45% of high-risk outpatients and 60% of inpatients now receive treatment on the same day as referral, compared with just 10% and 33% 4 years earlier.[13] This improved target attainment has been achieved through a variety of approaches, with some centres admitting more patients for assessment and treatment and others designing services with excess routine clinic capacity.[14]

Service reconfiguration remains ongoing in routine clinical practice, and it is unclear what impact such changes have on service costs and clinical outcomes. This study used discrete event simulation modelling to assess the impact of TIA service reconfiguration in two large urban hospitals with different approaches to service provision.

## METHODS
An extended methods section can be found in the online supplementary data.

### Setting and service design
The model was designed to assess the costs and clinical consequences arising from patients suspected of suffering an acute TIA, referred to two large urban hospitals in the West Midlands region of the UK. Both hospitals ran a specialist TIA outpatient service catering for approximately 500 patients with suspected TIA and minor stroke every year. The model parameters were defined by the characteristics of individuals attending these services, recruited to an observational study[15] during 2011 (table 1).

The availability of specialist TIA clinics was modelled on the basis of existing service provision at participating hospital sites during the study period. Patients could either be seen in traditional outpatient clinics, admitted, or seen on the ward on an outpatient basis. The original services were designed within the confines of available clinical staff and in particular a lack of specialist cover at the weekends. Services were redesigned at each site during the study period and the impact of these changes on cost and outcomes was examined by modelling both the original and modified service. The number and distribution of clinics in each service are described in table 2.

The impact of further hypothetical adjustments to the modified service at each site was modelled to replicate the following scenarios:
1. The addition of high-risk clinic slots for patients presenting on a Saturday and Sunday.
2. Including a weekend service but reducing the number of routine weekday clinics, by up to five clinic slots (one patient per slot) per week.

### Overview of the model
A discrete event simulation model was programmed in Delphi V.4 (Borland, San Francisco, CA, USA). It was adapted from a previously developed model and further details regarding the general modelling framework can

**Table 1** Patient characteristics as modelled

| Patient characteristics | |
| --- | --- |
| Age (mean±SD) (years) | 74±12 |
| Gender (female, %) | 240 (48.9) |
| Systolic blood pressure (mean±SD) (mm Hg) | |
| TIA mimic | 143±26 |
| TIA | 147±22 |
| Minor stroke | 144±18 |
| Overall mean | 145±24 |
| Final diagnosis (Actual condition) | |
| High-risk TIA | 116 (23.6%) |
| Low risk TIA | 46 (9.4%) |
| TIA mimic | 294 (60.0%) |
| Minor stroke | 34 (7.0%) |
| Patients referral to hospital via their GP according to diagnosis* | |
| High-risk TIA | 84 (72.3%) |
| Low-risk TIA/TIA mimic | 299 (87.9%) |
| Minor stroke | 20 (57.4%) |
| Overall mean | 402 (82.1%) |

*Expected numbers are rounded to the nearest integer, the apparent anomaly with the addition results from this rounding. High-risk patients defined as an ABCD$^2$ score of >4.
GP, general practitioner; TIA, transient ischaemic attack.

be found in the original report.[16] Briefly, the model generated a number of possible (virtual) patient histories, which began at the onset of TIA (or TIA-like) symptoms, and followed the patient along the clinical pathway from initial presentation to follow-up for subsequent stroke morbidity and mortality. An essential feature of the model was that patients were sharing limited resources in the form of routine clinics.

### Clinical pathways in the model
Following onset of an initial event, patients were assumed to contact either their general practitioner (GP) or attend the emergency department (ED) (online supplementary figure). The probability that a patient would choose a specific route was dependent on the type of patient (high/low risk), the time of day and day of week, estimated using data from the Oxford Vascular Study.[17] Following this initial contact with a healthcare professional, patients were referred to a specialist TIA outpatient clinic, seen on the ward (as an outpatient) or admitted as an inpatient. The type of referral was dependant on the specific hospital service provision, clinic availability and the level of risk of attending the patient, defined according to the ABCD$^2$ score.[10] During clinic attendance, it was assumed that the appropriate treatment would be initiated (ie, blood pressure lowering, cholesterol lowering or antiplatelet therapy in accordance with guidelines) and a small proportion of patients (4.1%) would be treated

**Table 2** Pattern of outpatient clinics for suspected transient ischaemic attack based on actual service provision at participating hospitals during patient recruitment period

| Hospital and service | | No of routine clinic slots available by day of the week | | | | | Total clinics per year | Details of clinic allocation within the model |
|---|---|---|---|---|---|---|---|---|
| | | Mon | Tues | Wed | Thur | Fri | | |
| Hospital 1 | Original service* | 2 | 4 | 4 | 4 | 2 | 724 | All patients are assigned to the next available clinic slots in order of referral. Where two referrals are made in 1 day, high-risk patients are given priority. |
| | Modified service* | 4 | 4 | 2 | 3 | 4 | 769 | Where two referrals are made in 1 day, high-risk patients are given priority. One slot is reserved at the end of each clinic for high-risk referrals. If the next high-risk clinic slot is not within 24 hours of referral, an additional slot is made available (up to one per day). All other patients are assigned to the next available clinic slots in order of referral. |
| Hospital 2 | Original service† | 6 | 4 | 2 | 0 | 0 | 624 | Patients are assigned to the next available outpatient clinic. Those high-risk patients who cannot be seen within 24 hours are admitted (including those presenting at weekends). |
| | Modified service† | 4 or 6‡ | 0 | 4 | 0 | 4 | 676 | All high-risk patients are seen on the ward as outpatients as required (including at weekends). Patients referred before 10:00 are seen at 17:00 on the same day, patients referred after 10:00 are seen at 10:00 on the following day. All low-risk patients seen at the next available clinic in order of referral. |

*Clinics are divided among four specialists, each of whom were absent for approximately 7 weeks a year (annual leave). These clinics are assumed not to take place if the specialist is absent.
†Specialists were absent for approximately 7 weeks a year (annual leave). All absent clinicians were replaced by a specialist from another site within the Trust.
‡There is a 50% probability each week that either four or six clinic slots will be available.

with carotid endarterectomy.[12][18] Patients were assumed to take this treatment as prescribed and gain the full benefit in terms of stroke prevention.

### Model population

All patients in the model were hypothetical, although they were based on data from real patients and hospitals. Overall rates of patient presentation were based on initial runs of the model and ensured a realistic distribution of final diagnoses, which represented the study sample, previous literature[14] and expert opinion (60% TIA mimics, 33% genuine TIA and 7% minor stroke). High-risk and low-risk TIAs were defined according to the ABCD$^2$ score.[10] The observed ratio of low-risk/high-risk TIA patients used in the base-case of the model was supplemented in sensitivity analyses by estimates derived from those reported in previous studies[10][19][20] and the experience of stroke physicians from centres across the UK (D Sims, G Ford and C Roffe, personal communication). The range of these estimates tested in this sensitivity analysis for impact on the model results was: 2.5:1 (high:low risk; base-case) to 1:1, 5:1 and 7:1.[10][19][20]

### Follow-up and risk of repeat events

Hypothetical patients remained in the model until 1 year from symptom onset, after which the increased risk of repeat event returns close to normal,[7][8] unless they died or suffered a non-fatal disabling stroke. No distinction was made in the model between fatal strokes and non-fatal disabling strokes: these were labelled as 'major strokes'. The risk of a repeat event (TIA or stroke) was dependent on the type of initial event (minor stroke, true TIA or mimic), the ABCD$^2$-based risk prediction of a subsequent event and other relevant risk factors, such as age, presence of atrial fibrillation and medication prescribed.[10][18][21–26] Following a minor (non-disabling) stroke, patients remained in the model, but with an additional risk of mortality that could be reduced by appropriate treatment. Additional deaths from this cause were estimated and labelled in the model outputs as 'post-stroke deaths'. Modelled outputs are, therefore, derived from risk profile of the hypothetical patients, adjusting for the effects of treatments which would start at varying times in the different scenarios. Risks were modelled using a Weibull distribution for the time to event which allowed for a substantially increased risk in the short term, followed by a decreasing risk over time.[9] Examples of the modelled risk for two patients (with high-risk and low-risk characteristics) are given in figure 1. The model was run 100 times for a total simulated time of 12 years in each run.

### Costs and outcomes

The model included costs of any GP visit (from presentation to referral), or transport by ambulance to the ED and

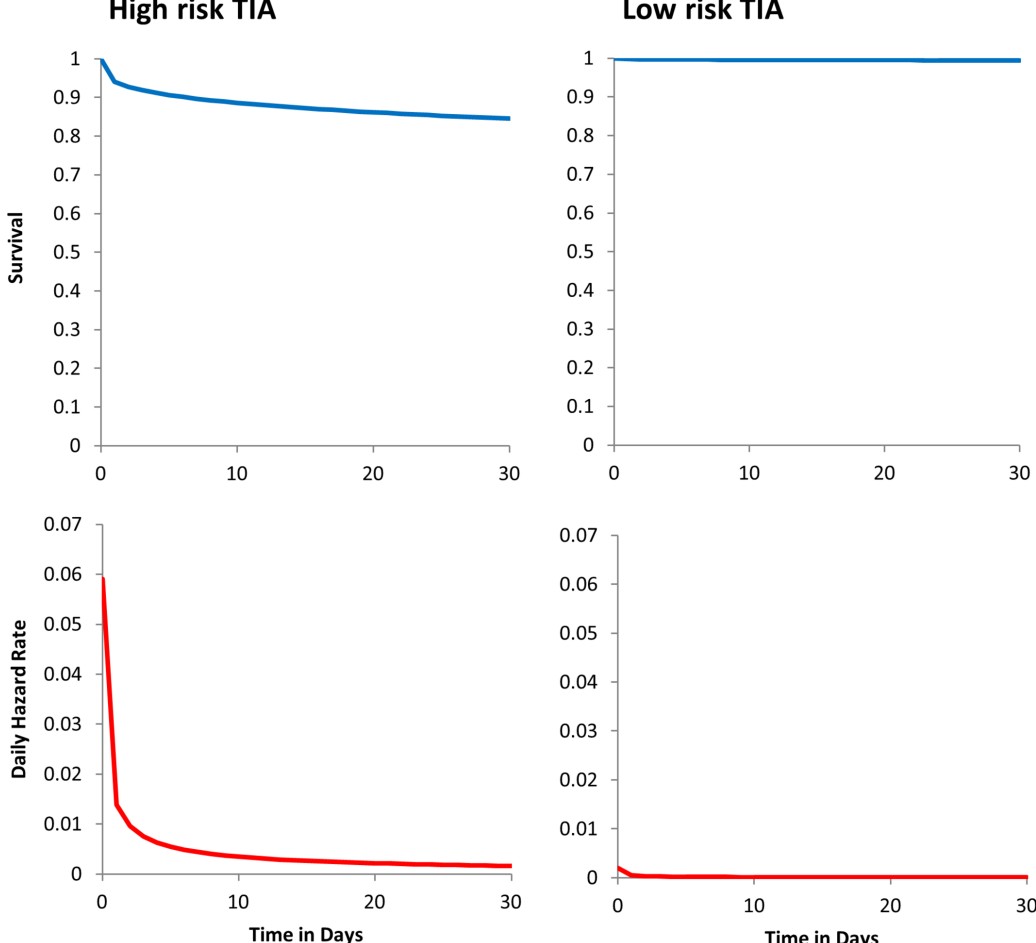

**Figure 1** Example survival curves and daily hazard rates for the risk of major stroke in high-risk and low-risk patients. High-risk patient: male age 70–74 years, SBP 156 mm Hg, speech disturbance without weakness, duration of symptoms 60+ min, not diabetic (resulting in ABCD and $ABCD^2$ scores of 5), undiagnosed Atrial fibrillation (AF), cholesterol 5.8. Low-risk patient: male age 70–74 years, SBP 115 mm Hg, speech disturbance without weakness, duration of symptoms 0–9 min, not diabetic (resulting in ABCD and $ABCD^2$ scores of 2), AF on warfarin, cholesterol 6.0.

ED attendance, outpatient clinics or hospital admission for TIA and stroke, surgery and therapy (online supplementary tables 1 and 2). Prehospital and treatment-related costs were the same for all options considered in this study since the focus was to compare comparative differences between modelled services caused by different clinic configurations. Costs included in the model were taken from a combination of NHS reference costs[27 28] and drug costs from the British National Formulary[29] and were modelled from a health services perspective. The price year for all costs was 2011–2012.

The primary outcome was the number of expected major strokes occurring post TIA, based on risk analysis. Secondary outcomes included the overall costs of service provision per year and attainment of national targets for TIA service provision. Target attainment was defined as the number of high-risk and low-risk 'breaches', which occurred in each service per year: high-risk breaches were defined as a high-risk patient not seen by a specialist within 24 hours of initial referral.[11 12] Low-risk breaches were defined as low-risk patients not seen by a specialist within 7 days of referral.[11 12] Further

outcomes examined the median time from referral to specialist appointment, the total number of routine outpatient appointments available (used or unused) and any unscheduled outpatient appointments required (where high-risk patients were assessed immediately on the ward).

All data are presented as means or medians±SD, IQR or 99% CI, chosen because multiple values are compared. Percentages are given for the total population unless otherwise stated.

## RESULTS

### TIA service usage

The model estimated a total of 490 patients would be referred to specialist TIA clinics each year at each hospital site (mean age 74±12 years; 48.9% female (table 1)). Of these, approximately 34 (7%) patients would be considered to have minor stroke and 162 (33%) would subsequently be considered to have confirmed TIA: the numbers classified as high/low risk, were 116/46, a ratio of 2.5:1. All subsequent results

**Table 3** Costs, resource utilisation and outcomes (per year) of modifying TIA service provision in hospital 1

| | Original service (16 clinic slots per week) | Modified service (17 clinic slots per week) | Weekend service+17 clinic slots per week) | Weekend service+15 clinic slots per week) | Weekend service+13 clinic slots per week) |
|---|---|---|---|---|---|
| Days operating per week | 5 | 5 | 7 | 7 | 7 |
| Total no of patients presenting | 491 | 490 | 490 | 491 | 491 |
| Cost of clinics used and unused | £340 000 | £361 000 | £366 000 | £346 000 | £325 000 |
| Major strokes post TIA (mean, 99% CI)* | 10.6 (10.4 to 10.8) | 10.7 (10.5 to 10.9) | 10.6 (10.4 to 10.8) | 10.8 (10.6 to 11.0) | 10.6 (10.4 to 10.8) |
| Poststroke deaths (mean, 99% CI)* | 3.0 (2.9 to 3.1) | 3.0 (2.9 to 3.1) | 3.0 (2.9 to 3.1) | 3.1 (3.0 to 3.2) | 3.0 (2.9 to 3.1) |
| No of high-risk breaches (%)† | 103 (69) | 32 (21%) | 6 (4%) | 7 (5%) | 8 (5%) |
| No of low-risk breaches (%)‡ | 7 (2) | 4 (1%) | 3 (1%) | 13 (4%) | 75 (22%) |
| Time from referral to clinic appointment for high-risk patients in days, median (IQR) | 1.15 (0.93–2.88) | 0.85 (0.17–0.99) | 0.68 (0.16–0.93) | 0.70 (0.15–0.93) | 0.86 (0.16–0.99) |

*Point estimate and 99% quasi CI reflecting the uncertainty from sampling in the model, not any uncertainty in model parameters. 99% was chosen because of multiple values were compared.
†High-risk breaches were defined as high-risk patients not seen by a specialist within 24 hours of initial clinic referral.[12]
‡Low-risk breaches were defined as low-risk patients not seen by a specialist within 7 days of initial clinic referral.
Of the ~490 patients in the model, 340 are considered low risk (294 TIA mimic; 46 low-risk TIA) and 150 are considered high-risk (116 high-risk TIA; 34 minor stroke).
TIA, transient ischaemic attack.

were robust to adjustment of this high-risk:low-risk ratio in the sensitivity analyses.

### Costs and outcomes of original TIA service provision
The baseline incidence of major strokes post TIA was 10.6 per year (99% CI 10.4 to 10.8) in hospital 1 and 10.7 per year (99% CI 10.5 to 10.9) in hospital 2. The cost of providing TIA services at both sites ranged from £340 000 to £368 000 per year (table 3 and online supplementary table 3). Services included the provision of 624–724 specialist week day clinic slots per year. In hospital 1, few high-risk patients were seen within 24 hours of referral (47/150 patients (31%)) and the median time from referral to clinic appointment was 1.15 days (IQR 0.93–2.88). In hospital 2, where all high-risk TIAs were admitted (a target attainment of 100%), the median time from referral to being seen and treated by a specialist was 0.85 days (IQR 0.74–0.94).

### Impact of service modification
Observed service modification increased the number of high-risk patients seen within 24 hours in hospital 1 (118/150; guideline target attainment of 79%), with increased service costs (increased from £344 000 to £366 000 per year) due to an increase in the number of unused routine clinic appointments (225–316 unused clinic slots per year) (figure 2). This increase in target

attainment resulted in a median time from referral to clinic appointment of 0.85 days (IQR 0.17–0.99); the incidence of major strokes post TIA remained largely unchanged (10.6 per year, 99% CI 10.4 to 10.8 (modified service) vs 10.7 per year, 99% CI 10.5 to 10.9 (original service)). Similar findings were observed at the second hospital site where costs were marginally reduced by not admitting patients with high-risk TIA (decrease from £368 000 to £363 000 per year) but there was no change in the number of patients being seen within 24 hours (150/150; guideline target attainment of 100%), median time from referral to clinic appointment (0.75 days, IQR 0.08–0.88 (modified service) vs 0.85 days, IQR 0.17–0.99 (original service)) or incidence of major strokes post TIA (10.4 per year, 99% CI 10.2 to 10.4 (modified service) vs 10.7 per year, 99% CI 10.5 to 10.9 (original service)).

### Potential optimal service provision
Modelling the impact of further hypothetical service reconfiguration, including a 7-day service, suggested that it was possible to reduce service costs and the number of unused clinic appointments and improve guideline target attainment, but this had little impact on the incidence of major stroke post TIA or poststroke deaths (table 3 and online supplementary table 3). Figure 2 shows the trade-off between

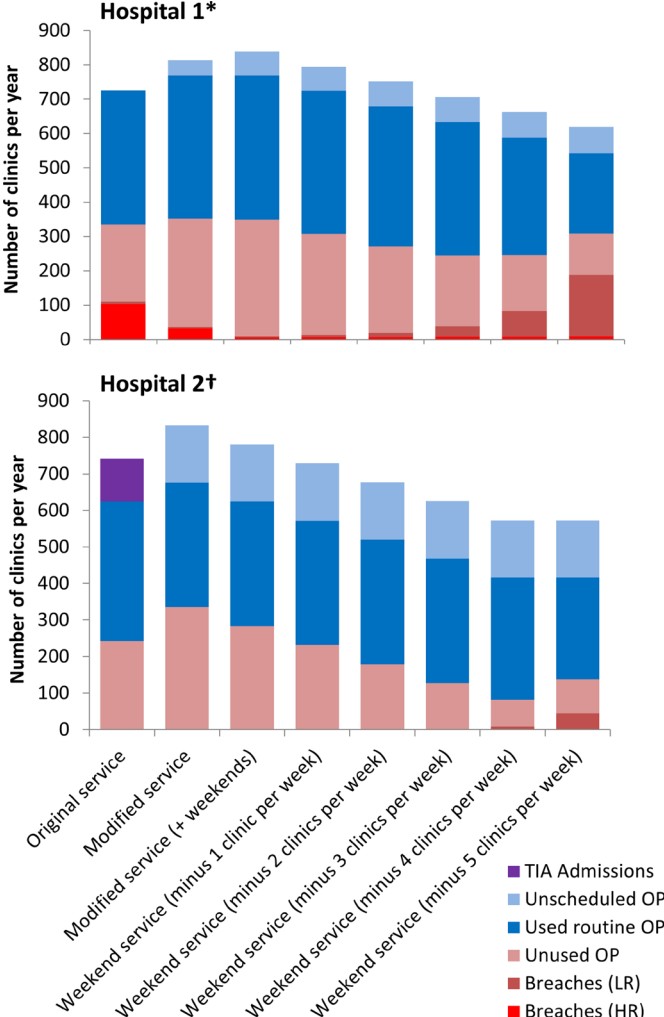

**Figure 2** Outcomes of service provision in terms of clinic appointment utilisation and guideline breaches. Original and modified services (first two bars at each hospital) were those which were actually implemented in each hospital. The remaining weekend services are hypothetical. *Hospital 1 included 16 routine clinics, 5 days per week (original service), 17 in the modified service and 19 in the modified service with weekend working. †Hospital 2 included 12 routine clinics, 3 days per week (original service) and 14 in the modified service (which also included weekend working). HR, high risk; LR, low risk; OP, outpatient appointment; TIA, transient ischaemic attack.

routine appointment provision, unscheduled appointments and high/low-risk guideline breaches. Assuming unused appointments (pink), unscheduled appointments (in light blue), admissions (purple) and breaches (red (high-risk breaches) and light red (low-risk breaches)) are preferably avoidable outcomes of service provision, the optimal hypothetical service in hospital 1 involved a weekend service with approximately 17–18 routine clinic appointments per week. In hospital 2, where more unscheduled appointments were incurred, the optimal hypothetical service involved a weekend service with 10–11 routine clinic slots distributed across the week on Mondays, Wednesdays and Fridays (with high-risk patients being seen in unscheduled appointments on the ward on any day of the week if necessary).

## DISCUSSION

The aim of this study was to model the impact of TIA service reconfiguration in two large urban hospitals and establish the impact on clinical outcomes (major strokes), costs and clinic usage. Reconfiguration of services was found to be effective and appropriate for reducing resource (clinic slot) waste, and improving guideline target attainment with variable effects on costs. However, in hospitals where services, despite poor target attainment, had minimal patient delays, such as those studied here, significant reconfiguration had little impact on modelled clinical outcomes such as major stroke post TIA. These findings suggest that clinicians and service coordinators should be cautious before initiating significant reconfiguration of services which are already seeing high-risk patients within 1–2 days, unless there is an obvious need to free up resources which could be used for another purpose (eg, managing other types of patients). While national targets for care are important, consideration should be given to revising them to maximise the benefits of attainment while not incentivising small changes that might increase costs for no measurable clinical benefit.

### Strengths and limitations

This study used local hospital data to describe the characteristics of the presenting population and thus the results are likely to be representative of the hospitals studied, but not necessarily generalisable to other centres in the UK and across the world. Despite this, guideline target attainment in hospital 1 (original service configuration) was similar to that reported nationally (31% vs 37% nationally),[13] which suggests that this hospital was more likely representative of centres across the UK.

Modelling allows estimation of service 'unknowns', such as the number of unscheduled clinic appointments required to meet guideline targets for high-risk patients. This requires preprogramming with hypothetical patients and specific rules; therefore, simplification of certain service intricacies is necessary. For example, the process by which high-risk patients were allocated unscheduled clinic appointments was formalised such that a patient referred by the ED physician was assumed to have been seen by a stroke specialist within 2 hours. However, patients referred by their GP were assumed to be seen by the stroke specialist on the ward at 10:00 the following morning or 17:00 the same day, whichever was later (but within 24 hours of initial referral). In reality, services are often adapted in response to a specific set of circumstances such as clinic room availability and thus the services modelled here, and the patients attending these services, can best be thought of as stylised versions of the actual services available and patients attending the hospitals in question.

The risk of recurrent stroke attributed to hypothetical patients in the model was based on an estimated ABCD[2] score.[10] The accuracy of this score for predicting

recurrent stroke has been called into question with a recent systematic review showing it to have poor discrimination between those at low risk and high risk of early (7 days) stroke.[30] However, this was the score used and recommended in practice at the time these services were modelled and was used for all service configurations examined, so our relative comparisons are likely to remain valid, even if the absolute numbers may be subject to change.

The costs modelled here were taken from standard NHS reference costs[27 28] and included investigations, treatment and the cost of staff attending patients during each appointment/admission. Costs relate to standard weekday services and increased rates for weekend services were unavailable for this analysis. It is, therefore, likely that the costs of providing a weekend service may have been underestimated in the present analyses.

### Findings in the context of previous literature and implications for practice

Economic modelling has previously been used for determining the impact of service reconfiguration in acute stroke[31–34] but studies examining optimal strategies for TIA service provision are less common. Those that exist, focus on whether patients should be admitted or be seen as an outpatient.[16 35 36] It is generally accepted that patient admission for TIA is inefficient and not cost-effective,[16 37–39] but there are a paucity of objective data comparing different configurations of outpatient service in the same setting, limited to questionnaire surveys of stroke physicians[14 40] and the Sentinel Stroke National Audit Programme (SSNAP) in the UK.[13] A recent review identified six models of care designed to avoid patient admission for TIA. These ranged from patient assessment and triage in the ED, a 24 hour nurse-led telephone advice service and the primary care triage system studied here (+/−electronic decision support).[41] The appropriate model of care will depend on the local healthcare system, but consideration of how such a model is implemented in terms of service configuration is warranted, regardless of the system used.

The primary finding of this study was that while service reconfiguration can significantly improve guideline target attainment and potentially reduce service costs, such changes have little impact on the estimated prevalence of subsequent stroke, when services are already working efficiently. In this study, even the largest service reconfiguration (introduction of a 7-day service) only reduced the median time from patient referral to clinic attendance and treatment by 0.3 days (8 hours). This enabled a large number of patients to be seen within 24 hours, improving guideline target attainment but had little impact on stroke risk. The recommendations for a 'see and treat within 24 hours' target for high-risk patients were originally based on data from the EXPRESS study,[9] which showed that those patients seen and treated within 24 hours of initial referral had significantly better outcomes than those patients who were not. However, in the original service to which this optimal strategy was compared, the average time from referral to clinic appointment was 3 days (IQR 2–5 days) and time to first treatment was 20 days (IQR 8–53). The improvement in time to initiation of treatment was therefore 20 days, compared with an improvement of 8 hours seen in this study, which explains the very different results in terms of clinical outcome.

Recent results from SSNAP show that 45% of high-risk outpatients and 60% of high-risk inpatients are seen and treated within 24 hours of referral, suggesting that there is plenty of room for improvement. However, with a median time to first clinic appointment (and treatment) for all patients of just 2 days, the present analyses would suggest that further improvements in guideline target attainment may have little impact on major stroke following TIA.

The modified services examined and proposed here may require the capacity to accommodate unscheduled clinic appointments, which is likely to have an opportunity cost affecting the quality of other services running concurrently in the hospital and may result in increased costs with little tangible benefit. In particular, the ability to accommodate such appointments depends on the time of day at which the patient presents, the availability of specialist stroke physicians and scanning equipment required to make an accurate diagnosis, as well as the needs of competing services within the hospital (eg, acute stroke services).

### CONCLUSIONS

Reconfiguration of specialist services for TIA can be effective and appropriate for reducing costs, reducing the number of unused routine clinic appointments and increasing clinical guideline target attainment. However, where services are already working near optimal, such modification has little impact on clinical outcomes such as subsequent stroke. Clinicians and service coordinators should be cautious before initiating significant reconfiguration of services which are already seeing high-risk patients within 1–2 days, although this might still be appropriate if resources within the hospital are freed up and could be used for an alternate purpose.

**Author affiliations**

[1]Health Economics Unit, University of Birmingham, Birmingham, UK
[2]Nuffield Department of Primary Care Health Sciences, NIHR School for Primary Care Research, University of Oxford, Oxford, UK
[3]Oxford Academic Health Science Network, Oxford University Hospitals NHS Foundation Trust, Oxford, UK
[4]Nuffield Department of Medicine, University of Oxford, Oxford, UK
[5]Primary Care Unit, University of Cambridge, Cambridge, UK
[6]Department of Public Health, NHS Lanarkshire, Bothwell, UK
[7]Centre for Health and Social Care Research, Faculty of Health, Social Care and Education, St George's University of London, Kingston University, London, UK
[8]Nuffield Department of Clinical Neurosciences, University of Oxford, Oxford, UK
[9]Geriatric Medicine, Heart of England NHS Foundation Trust, Birmingham, UK
[10]Stroke Medicine, University Hospitals Birmingham NHS Foundation Trust, Birmingham, UK

**Acknowledgements** We would like to thank Amun Boyal, Kelly Rosborough and the staff at both participating hospitals for their assistance with recruitment and data collection. Thanks to Sheila Bailey and Anita Martin for their administrative support and Primary Care Clinical Research and Trials Unit for building and maintaining the study database.

**Collaborators** Birmingham and Black Country Collaborations for Leadership in Applied Health Research and Care investigators include: Peter Carr, Heart of England NHS Foundation Trust; Sheila Greenfield, Primary Care Clinical Sciences, University of Birmingham; Brin Helliwell, Lay member of Steering Group, Christina Nand, Lay member of Steering Group; Norman Phillips, Lay member of Steering Group; Rob Scott, Birmingham and Midland Eye Centre; Satinder Singh, Primary Care Clinical Sciences, University of Birmingham; Matthew Ward, West Midlands Ambulance Service NHS Trust.

**Contributors** RJMcM, JM, SJ and PB had the original idea and gained the funding for the project. PB undertook the economic analyses. JPS and RMM were responsible for the data collection and JPS wrote the first draft of the manuscript with RJMcM and PB. All authors contributed to protocol development, refined the manuscript and approved the final version. RJMcM is the guarantor.

**Funding** This work was supported by the National Institute for Health Research (NIHR) as part of the Collaborations for Leadership in Applied Health Research and Care (CLAHRC) programme for Birmingham and Black Country. JPS held a Medical Research Council (MRC grant number MR/K022032/1) Strategic Skills Postdoctoral Fellowship and is now funded by the NIHR Oxford CLARHC. RJMcM holds an NIHR Professorship.

**Competing interests** None declared.

**Ethics approval** Approval for this project was obtained from the National Research Ethics Service (NRES) Committee, London – Queen Square (reference; 09/H0716/71).

**Provenance and peer review** Not commissioned; externally peer reviewed.

**Data sharing statement** The datasets analysed during the current study are available from the corresponding author on reasonable request.

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
