## [Reviewer comments · BMJ Open]

ARTICLE DETAILS

TITLE (PROVISIONAL)	When has service provision for transient ischemic attack improved enough? A discrete event simulation economic modelling study
AUTHORS	Barton, Pelham; Sheppard, James; Penaloza, Cristina; Jowett, Sue; Ford, Gary; Lasserson, Daniel; Mant, Jonathan; Mellor, Ruth; Quinn, Tom; Rothwell, Peter; Sandler, David; Sims, Don; McManus, Richard

VERSION 1 – REVIEW

REVIEWER	Dominique Cadilhac Monash university, Australia No Competing Interest
REVIEW RETURNED	07-Jul-2017

GENERAL COMMENTS	I found this to be an interesting and well written paper. I did identify that some more recent literature from this area was not used and provide the references and other feedback in the attached file. In the ‘Strengths and limitations of this study’ section (page 4) the authors claim this is the ‘first’ study to model impact of routine service provision for patients with TIA. Please see these other papers and amend accordingly. You may consider how these other papers might be used in other parts of your paper e.g. Discussion Sanders L, Cadilhac DA, Srikanth V, Pei Chong C, Phan TG. Is nonadmission-based care for TIA patients cost-effective? A microcosting study. Neurology: Clinical Practice 2015; 5(1):58-66, doi 10.1212/CPJ.0000000000000095 Buisman LR, Tan SS, Nederkoorn PJ, Koudstaal PJ, Redekop WK. Hospital costs of ischemic stroke and TIA in the Netherlands. Neurology 2015; 84(22): 2208-15. Ranta A, Dovey S, Weatherall M, O’Dea D, Gommans J, Tilyard M. Cluster randomized controlled trial of TIA electronic decision support in primary care. Neurology 2015; 84(15): 1545-51. Ranta A, Lichtman JH. Health economics of cerebrovascular disease: Can we do better? Neurology 2015; 84(22): 2204-5. Edits for the Abstract page 3. Design: (line 5) — please add the ref year Interventions: (line 8) — The model examined [delete and insert “included”] (line 8) — .previous and modified services [insert ‘existing versus’ before ‘previous and modified services’]
--

	Outcome measures: (line 10) — primary outcome was the prevalence of disabling stroke. How were deaths captured or explain why excluded? (line 13) — high risk patients seen within 24 hours [how defined?] Results: (line 14) — 490 patients – Please add in some basic characteristics of the cohort e.g. % male, median age, % high risk as well as the ref year for costs. (line 18) £5000— £21,000 [are these boot strapped median costs?] Conclusions: (line 24) — estimated impact on clinical outcomes [so extra spend not worthwhile?] Page 4 (Strengths and limitations of this study) (lines 3 and 4) – Priority claim (see feedback above) - Did you mean in UK? (lines 7 to 9) — Costs may be underestimated, and also doesn't take into account implications of implementation. (line 11) — Would penalty rates for weekend not be known so could have been included? Page 5 Background (line 11) — maybe better [insert space: may be] Page 6 (Setting and service design) (line 13) — subsequently underwent redesign – ? In reality or modelled assumptions might need to clarify for readers more explicitly (line 21) — clinic slots per week: how many patients seen in a slot? Page 7 (Clinical pathways in the model) (lines 14 to 17) — Did you consider patient adherence to prevention therapies? Page 8 (Costs and outcomes) (line 22) — How were deaths included/excluded? Some patients will die after a major stroke event subsequent to TIA. If excluded why? Page 9 (Results) (line 12) – 490 patients, please add in some basic characteristics of the cohort [mean age, % male, etc). How was the 490 derived was there a range for patient numbers or fixed? Does this number of modelled cases accord with what happens now at these hospitals in terms of case load? Page 11 (Discussion) (line 15) — ‘obvious potential to reduce costs – [unsure if this is the right wording - alternatively these freed up resources could be used for another purpose e.g. managing other types of patients] Page 12 (line 17) – ‘...be increased by provision of services at the weekend’ - [any staff penalty rates in the UK? Use these to approximate] – I think if penalty rates are paid then weekend services were underestimated rather than saying ‘possibly’. (line 24) — Sanders L et al Neurology: Clinical Practice 2015; 5(1):58-66 paper not referenced and may have been missed in author literature review Page 14 (Conclusions) (line 16) – Might still be appropriate if resources within the hospital
--	--

	are freed up and can be used for an alternate purpose Page 24 (Table 2) In reality could these be scaled up, which other aspect of hospital service would miss out? Page 30 (Extended methods) (line 25) — underwent redesign – how/who worked out the features? What was the process of consensus? Page 35 (eTable 1) (line 6) Source – and he experience – [fix “and the experience”] Define ABCD2 and GP in footnotes as well as SDP ONS Page 37 (eTable 3) (line 17 to 19) – number of high risk breaches and number of low risk breaches – 1st column original service (12 clinic slots per week) - 0(0.0%) [Is this correct?]
--	---

REVIEWER	Anna Ranta University of Otago, Wellington, New Zealand No Competing Interest
REVIEW RETURNED	09-Jul-2017

GENERAL COMMENTS	Thank you for the opportunity to review this interesting manuscript. This is novel work that will be of great interest to stroke clinicians and service managers and I would recommend this for publication. I have just a few, mostly minor comments for the authors to consider (to clarify, I have used the journal provided line numbers rather than those included by the authors). I have ticked 'no' in two instances above. Both issues are readily addressed by responding to the questions listed below (either by accepting my suggestions or rebutting them). No major revisions are required in my mind. Abstract and Background: no comments Methods (1) page 6 line 29 - "Both services subsequently underwent redesign" - it would be good to add a bit more detail here as to what the exact changes were; I realise there is more detail in the supplement, but most will not read this a a couple more sentences would be of value in the main paper. (2) page 7 line 1 - need to clarify: so no actual patients were used for this study? all hypothetical, correct? I think this could be further emphasised especially as most readers will be somewhat unfamiliar which such modelling work compared with clinical studies looking at actual patients. (3) page 7 line 31 - is it just ABCD2? not also crescendo TIAs, other high risk patients? (4) page 7 line 51 - so risk of stroke is all based on ABCD2 data? this has of course been contested (worth mentioning this in the discussion) and if not then this could perhaps do with a bit more detail. (5) page 8 line 5 - not sure the word 'patient' is right...'hypothetical patients?'- just still thinking of clinicians reading this used to studies
---

	involving actual patients. (6) page 8 line 49 - number of disabling strokes per model...again these are not actual patients with actual strokes if I understand correctly so would perhaps rephrase as 'number of expected strokes based on risk analysis/' Results (7) page 9 line 40 - are these actual baseline stroke rates for these hospitals? that's pretty high....would be good to reference this or is this based on the risk profile of hypothetical patients based on published ABCD2 stroke risk data? All of this could do with just a bit more detail in the main manuscript I think. (8) page 9 line 44 - could average or median cost be depicted per patient with a SD/IQR? Overall it seems to be a pretty low per patient cost of only about 1000/pt or so? that includes admissions and all the investigations, medications, and doctor visits for a year? Discussion (9) page 11 Strengths and limitations - seems like the limitations around not having actually followed real patients could be emphasized a bit further. (10) page 12 line 23 - I was surprised that the authors do not reference the recent summary paper on internationally available TIA services published in Neurology in early 2016 (http://www.neurology.org/content/early/2016/01/22/WNL.0000000000002339.abstract). This would help give a bit of a broader perspective than just the UK viewpoint. Some of the reviewed papers in that publication also include health economic data that may be of interest to the authors/relevance to this paper.
--	--

VERSION 1 – AUTHOR RESPONSE

Reviewer 1 comments

(1) In the 'Strengths and limitations of this study' section (page 4) the authors claim this is the 'first' study to model impact of routine service provision for patients with TIA. Please see these other papers and amend accordingly. You may consider how these other papers might be used in other parts of your paper e.g. Discussion

Response: We thank the reviewer for highlighting these references and have added them to the discussion section of our paper. The only exception is the Ranta et al., decision support trial, which whilst very interesting, examines the most appropriate method of patient referral to specialist care, rather than the question examined here – namely what is the optimal configuration of TIA outpatient services. We have changed the strengths and limitations section on page 4 to reflect this:

“This is the first study to model the impact of different outpatient service provision for patients suffering transient ischemic attack.”

(2) Edits for the Abstract page 3.

Design: (line 5) — please add the ref year

Response: We have added this.

“Design: Discrete event simulation model using data from routine electronic health records from 2011.”

(3) Interventions: (line 8) — The model examined [delete and insert “included”] (line 8) — .previous and modified services [insert ‘existing versus’ before ‘previous and modified services’]

Response: We have corrected this:

“Interventions: Included existing versus previous (less same day clinics) and hypothetical service reconfiguration”

(4) Outcome measures: (line 10) — primary outcome was the prevalence of disabling stroke. How were deaths captured or explain why excluded?

Response: This figure actually includes fatal strokes as well as non-fatal disabling strokes. We have changed the term "disabling" to "major" to reflect this and defined this better in the methods. We also modelled a heightened risk of death following a minor stroke, as explained in the edited methods section:

“Hypothetical patients remained in the model until one year from symptom onset, after which the increased risk of repeat event returns close to normal, unless they died or suffered a non-fatal disabling stroke. No distinction was made in the model between fatal strokes and non-fatal disabling strokes: these were labelled "major strokes". The risk of a repeat event (TIA or stroke) was dependent on the initial event (minor stroke, true TIA or mimic), the severity of that event (measured by ABCD2 score) and other relevant risk factors such as age, presence of atrial fibrillation and medication prescribed. Following a minor (non-disabling) stroke, patients remained in the model, but with an additional risk of mortality that could be reduced by appropriate treatment. Additional deaths from this cause were estimated and labelled in the model outputs as "post-stroke deaths". Modelled outputs are therefore derived from risk profile of the hypothetical patients, adjusting for the effects of treatments which would start at varying times in the different scenarios.”

We have added a row to each of the relevant results tables to account for this.

There is very little variation in the post stroke deaths in the different models: this has been noted in the discussion, where we now say on Page 12

"little impact on ... stroke post TIA or post-stroke deaths"

(5) (line 13) — high risk patients seen within 24 hours [how defined?]

Response: Risk was defined according to ABCD2 score; we have added this to the abstract:

“...proportion of high risk patients [according to ABCD2 score] seen within 24 hours).”

(6) Results: (line 14) — 490 patients – Please add in some basic characteristics of the cohort e.g. % male, median age, % high risk as well as the ref year for costs.

These have now been added:

“The estimated costs of previous service provision for 490 patients (aged 74±12 years, 48.9% female and 23.6% high risk)....”

(7) (line 18) -£5000 to +£21,000 [are these boot strapped median costs?]

Response: No. These figures represent the differences in costs for the modified service compared to the original at each modelled site. In Hospital 1 cost up by £21,000 (see Table 3) while in Hospital 2, costs went down by £5000 (eTable 3). We have amended this sentence to clarify this:
“The costs associated with the existing and hypothetical services decreased by £5,000 at one site and increased by £21,000 at the other site.....”

(8) Conclusions: (line 24) — estimated impact on clinical outcomes [so extra spend not worthwhile?]

Response: We have adjusted this:
“it has little estimated impact on clinical outcomes and increased investment may not be worthwhile.”

(9) Page 4 (Strengths and limitations of this study) (lines 3 and 4) – Priority claim (see feedback above) - Did you mean in UK?

As mentioned above, the main reason our study is unique that it is the first to compare different models of outpatient service provision, rather than outpatient vs. inpatient services. We have amended this to make it clearer:

“This is the first study to model the impact of different outpatient routine service provision for patients suffering transient ischemic attack.”

(10) (lines 7 to 9) — Costs may be underestimated, and also doesn't take into account implications of implementation. (line 11) — Would penalty rates for weekend not be known so could have been included?

Response: Unfortunately increased rates for weekend services were unavailable for this analysis so could not be included. We have added the following additional point to the ‘strengths and weaknesses’ section on page 5:

“The potential costs of implementation of new services, for example those for weekend services, were unavailable and so the costs presented here may have been underestimated.”

(11) Background (line 11) — maybe better [insert space: may be]

Response: This has been added.

(12) (Setting and service design) (line 13) — subsequently underwent redesign – ? In reality or modelled assumptions might need to clarify for readers more explicitly (line 21) — clinic slots per week: how many patients seen in a slot?

Response: We have clarified these points and amended the text to read:
“Services were redesigned at each site during the study period....”
“.....by up to 5 clinic slots (one patient per slot) per week.”

(13) Page 7 (Clinical pathways in the model) (lines 14 to 17) — Did you consider patient adherence to prevention therapies?

Response: This study was focused more on the appropriateness of service design, rather than the effectiveness of treatment and so patients were assumed take the medications as prescribed and

gain the full benefit in terms of stroke prevention. To clarify this, we have added the following sentence:

“During clinic attendance, it was assumed that the appropriate treatment would be initiated Patients were assumed to take this treatment as prescribed and gain the full benefit in terms of stroke prevention.”

(14) Page 8 (Costs and outcomes) (line 22) — How were deaths included/excluded? Some patients will die after a major stroke event subsequent to TIA. If excluded why?

Response: We have added the death post stroke outcome to table 3 and eTable3 and referred to it in the methods and results as described above.

(15) Page 9 (Results) (line 12) – 490 patients, please add in some basic characteristics of the cohort [mean age, % male, etc). How was the 490 derived was there a range for patient numbers or fixed? Does this number of modelled cases accord with what happens now at these hospitals in terms of case load?

Response: We have added some basic population characteristics and referred to table 1 which provides more detail:

“The model estimated a total of 490 patients would be referred to specialist TIA clinics each year at each hospital site (mean age 74±12 years; 48.9% female [table 1]).”

Due to the way in which attendance at TIA clinics was recorded in participating hospitals, it was not possible to accurately define the case load of each site per year during the study period.

(16) Page 11 (Discussion) (line 15) — ‘obvious potential to reduce costs – [unsure if this is the right wording - alternatively these freed up resources could be used for another purpose e.g. managing other types of patients]

Response: We agree and have adjusted this as suggested:

“These findings suggest that clinicians and service coordinators should be cautious before initiating significant reconfiguration of services which are already seeing high risk patients within 1-2 days, unless there is an obvious need to free up resources which could be used for another purpose (e.g. managing other types of patients).”

(17) Page 12 (line 17) – ‘...be increased by provision of services at the weekend’ - [any staff penalty rates in the UK? Use these to approximate] – I think if penalty rates are paid then weekend services were underestimated rather than saying ‘possibly’.

As mentioned in response to point 10, penalty rates for weekend services were unavailable for this analysis so could not be included.

Response: We adjusted the limitations section of the discussion on page 14 to be more explicit about this:

“Costs relate to standard weekday services and increased rates for weekend services were unavailable for this analysis. It is therefore likely that the costs of providing a weekend service may have been underestimated in the present analyses.”

(18) (line 24) — Sanders L et al Neurology: Clinical Practice 2015; 5(1):58-66 paper not referenced and may have been missed in author literature review

Response: We have added this reference to the discussion section. It is worth noting that the Sanders paper is not directly comparable to this study in that it compares an admission based system to an outpatient service like previous studies we have already referenced. Our study is unique in its comparison of different outpatient service designs, as highlighted in the strength and limitations on page 4, and discussion section on page 14 where we say:

“.....studies examining optimal strategies for TIA service provision are less common. Those that exist, focus on whether or not patients should be admitted or be seen as an outpatient. It is generally accepted that patient admission for TIA is inefficient and not cost-effective, but there are a paucity of objective data comparing different configurations of outpatient service....”

(19) Page 14 (Conclusions) (line 16) – Might still be appropriate if resources within the hospital are freed up and can be used for an alternate purpose

Response: We agree although it is difficult to speculate where such resource savings might be made. We have revised this paragraph:

“Clinicians and service coordinators should be cautious before initiating significant reconfiguration of services which are already seeing high risk patients within 1-2 days, although this might still be appropriate if resources within the hospital are freed up and could be used for an alternate purpose.

(20) Page 24 (Table 2) In reality could these be scaled up, which other aspect of hospital service would miss out?

Response: As above, it is probably beyond the scope of this modelling study to speculate as to where else in the hospital resource savings might be made. Such decisions will be made at a local level and likely vary across sites and hospital Trusts. Our model therefore assumes that TIA services could be increased through an increase in budget.

(21) Page 30 (Extended methods) (line 25) — underwent redesign – how/who worked out the features? What was the process of consensus?

Response: All TIA services provided by each hospital were designed by the stroke teams responsible for delivering them.

“Both services were subsequently redesigned by the hospital stroke team responsible for delivering the service.”

(22) Page 35 (eTable 1) (line 6) Source – and he experience – [fix “and the experience”]
Define ABCD2 and GP in footnotes as well as SDP ONS]

This has been updated.

(23) Page 37 (eTable 3) (line 17 to 19) – number of high risk breaches and number of low risk breaches – 1st column original service (12 clinic slots per week) - 0(0.0%) [Is this correct?]

Response: This is not a mistake. As you will note from table 2, Hospital 2 saw all high risk patients immediately on the ward in their modified service (i.e. they did not need to be booked into a pre-specified slot), thus no high risk breaches occurred in this or any new service modelled. We have added a comment to this effect in the footnotes to the table.

Reviewer 2 comments

Methods

(1) Page 6 line 29 - "Both services subsequently underwent redesign" - it would be good to add a bit more detail here as to what the exact changes were; I realise there is more detail in the supplement, but most will not read this a a couple more sentences would be of value in the main paper.

Response: Details of the services and what was changed is given in table 2. We do not feel it is appropriate to repeat this in the text.

(2) page 7 line 1 - need to clarify: so no actual patients were used for this study? all hypothetical, correct? I think this could be further emphasised especially as most readers will be somewhat unfamiliar which such modelling work compared with clinical studies looking at actual patients.

Response: We agree and have added the following sentence to clarify this:

"All patients in the model were hypothetical, although they we based on data from real patients and hospitals."

(3) page 7 line 31 - is it just ABCD2? not also crescendo TIAs, other high risk patients?

Response: High and low risk TIAs were defined according to the ABCD2 score only, which at the time was the primary way of defining risk used by referring GPs and recommended in guidelines. We state this in the online supplement and have now added this to the main manuscript:

"High and low risk TIAs were defined according to the ABCD2 score."

We took into account the experience of stroke physicians from centres across the UK [personal communications, D Sims, G Ford and C Roffe] but because modelling other types of high risk TIA would require the need to collect and model further (potentially uncommon) patient characteristics, it was decided that such additional complexity was not worthwhile given that it would be unlikely to significantly affect the results.

(4) page 7 line 51 - so risk of stroke is all based on ABCD2 data? this has of course been contested (worth mentioning this in the discussion) and if not then this could perhaps do with a bit more detail.

Response: We note this and have added the following paragraph highlighting this issue in the strength and weaknesses section of the discussion:

"The risk of recurrent stroke attributed to hypothetical patients in the model was based on an estimated ABCD2 score.¹⁰ The accuracy of this score for predicting recurrent stroke has been called into question with a recent systematic review showing it to have poor discrimination between those at low and high risk of early (7 day) stroke.³⁰ However, this was the score used and recommended in practice at the time these services were modelled and was used for all service configurations examined, so our relative comparisons are likely to remain valid, even if the absolute numbers may be subject to change."

(5) page 8 line 5 - not sure the word 'patient' is right...'hypothetical patients?'- just still thinking of clinicians reading this used to studies involving actual patients.

Response: We agree and this has been updated.

(6) page 8 line 49 - number of disabling strokes per model...again these are not actual patients with actual strokes if I understand correctly so would perhaps rephrase as 'number of expected strokes based on risk analysis/'

Response: We agree and have updated the paper with the following:

"The primary outcome was the number of expected major strokes occurring post TIA, based on risk analysis." (We have changed "disabling" to "major" as noted in our response to comment (4) from Reviewer 1.)

(7) page 9 line 40 - are these actual baseline stroke rates for these hospitals? that's pretty high....would be good to reference this or is this based on the risk profile of hypothetical patients based on published ABCD2 stroke risk data? All of this could do with just a bit more detail in the main manuscript I think.

Response: These are model outputs derived from the risk profile of the hypothetical patients, adjusting for the effects of treatments which would start at varying times in the different scenarios. We have added the following sentence to the methods sections of the manuscript (page 9):

"The risk of a repeat event (TIA or stroke) was dependent on the initial event (minor stroke, true TIA or mimic), the severity of that event (measured by ABCD2 score) and other relevant risk factors such as age, presence of atrial fibrillation and medication prescribed. Following a minor (non-disabling) stroke, patients remained in the model, but with an additional risk of mortality that could be reduced by appropriate treatment. Additional deaths from this cause were estimated and labelled in the model outputs as "post-stroke deaths". Modelled outputs are therefore derived from risk profile of the hypothetical patients, adjusting for the effects of treatments which would start at varying times in the different scenarios."

(8) page 9 line 44 - could average or median cost be depicted per patient with a SD/IQR? Overall it seems to be a pretty low per patient cost of only about 1000/pt or so? that includes admissions and all the investigations, medications, and doctor visits for a year?

Response: We decided that it is not appropriate to present costs per patient since some of the costs incurred are not attributable to specific patients. The costs included here are the costs of running the clinic over a year, but each patient is only in the system for a few days.

(9) page 11 Strengths and limitations - seems like the limitations around not having actually followed real patients could be emphasised a bit further.

Response: As noted previously, these are hypothetical patients, based on the characteristics of real patients attending services at the hospitals studied. We have revised the following sentences in the strength and limitations section to highlight this:

"Modelling allows estimation of service 'unknowns', such as the number of unscheduled clinic appointments required to meet guideline targets for high risk patients. This requires pre-programming with hypothetical patients and specific rules;"

"In reality, services are often adapted in response to a specific set of circumstances such as clinic room availability and thus the services modelled here, and the patients attending these services, can

best be thought of as stylised versions of the actual services available and patients attending the hospitals in question.”

(10) page 12 line 23 - I was surprised that the authors do not reference the recent summary paper on internationally available TIA services published in Neurology in early 2016 (<http://www.neurology.org/content/early/2016/01/22/WNL.0000000000002339.abstract>). This would help give a bit of a broader perspective than just the UK viewpoint. Some of the reviewed papers in that publication also include health economic data that may be of interest to the authors/relevance to this paper.

Response: We thank the review for highlighting this reference which is very helpful. We have added the following to the discussion to highlight this:

“A recent review identified six models of care designed to avoid patient admission for TIA. These ranged from patient assessment and triage in the ED, a 24-hour nurse-led telephone advice service and the primary care triage system studied here (+/- electronic decision support).⁴¹ The appropriate model of care will depend on the local healthcare system, but consideration of how such a model is implemented in terms of service configuration is warranted, regardless of the system used.”

VERSION 2 – REVIEW

REVIEWER	Anna Ranta University of Otago
REVIEW RETURNED	18-Sep-2017

GENERAL COMMENTS	The reviewers have largely addressed my concerns and I feel the manuscript is improved. Thank you. I just have three minor comments for their further consideration. (1) The additional sentence referred to in the second point seems grammatically incorrect and may need editing: “All patients in the model were hypothetical, although they we based on data from real patients and hospitals.” should that read 'were' instead of 'we'? (2) Issue addressed in point seven. ABCD2 is not score of TIA severity and I would reword this. I also suggest adding 'type of' in front of second mention of the word 'event' - so suggest this wording: “..was dependent on the type of initial event (minor stroke, true TIA, or mimic), the ABCD2 based risk prediction of a subsequent event' and...” (3) I am still puzzled by the cost analysis. If we are only talking about clinic cost then why bother mentioning inpt costs etc. Also I continue to have some concerns about the cost estimates (400 pounds for admission and 4000 for CEA seem very low). However, I note that the figure of 1000/patient has been removed and instead the focus is on total clinic cost (although I didn't see any track changes?) which de-emphasizes the cost model used here and focuses the reader on relative cost differences between services so that's ok. Just please make sure the 1000/pt has indeed been taken out and consider rewording the method section to ensure it aligns with what is reported under results. Thanks
---

VERSION 2 – AUTHOR RESPONSE

Reviewer: 2

(1) The additional sentence referred to in the second point seems grammatically incorrect and may need editing: "All patients in the model were hypothetical, although they we based on data from real patients and hospitals." should that read 'were' instead of 'we'?

Response: The reviewer is correct, this was a typo which has now been corrected.

(2) Issue addressed in point seven. ABCD2 is not score of TIA severity and I would reword this. I also suggest adding 'type of' in front of second mention of the word 'event' - so suggest this wording: "...was dependent on the type of initial event (minor stroke, true TIA, or mimic), the ABCD2 based risk prediction of a subsequent event' and..."

Response: We agree with this clarification and this has now been amended in the manuscript.

(3) I am still puzzled by the cost analysis. If we are only talking about clinic cost then why bother mentioning inpt costs etc. Also I continue to have some concerns about the cost estimates (400 pounds for admission and 4000 for CEA seem very low). However, I note that the figure of 1000/patient has been removed and instead the focus is on total clinic cost (although I didn't see any track changes?) which de-emphasizes the cost model used here and focuses the reader on relative cost differences between services so that's ok. Just please make sure the 1000/pt has indeed been taken out and consider rewording the method section to ensure it aligns with what is reported under results.

Response: We apologise for the confusion which seems to be caused by our response to this question in our previous rebuttal. The costs included in the model relate to costs of running the clinic over a year and costs incurred prior to that at initial presentation: ie during initial GP clinic or the costs of transport by ambulance to the ED and ED attendance, prior to TIA clinic attendance (+/- any investigations undertaken or medications prescribed at the subsequent clinic). Pre-hospital/ treatment costs are the same for all options considered in this study and the comparative differences between modelled services occur due to the different clinic configurations.

At no point have we ever included per patient costs since the costs incurred are not attributable to specific patients. As the reviewer rightly states, the model is focused on the relative cost differences between services. We have clarified this in the methods section of the manuscript with the following:

"The model included costs of any GP visit (from presentation to referral), or transport by ambulance to the ED and ED attendance, outpatient clinics or hospital admission for TIA and stroke, surgery and therapy (eTables 1 and 2, online data supplement). Pre-hospital and treatment related costs were the same for all options considered in this study since the focus was to compare comparative differences between modelled services caused by different clinic configurations."